# Investigating the functional and structural effect of non-synonymous single nucleotide polymorphisms in the cytotoxic T-lymphocyte antigen-4 gene: An *in-silico* study

Md. Mostafa Kamal[1], Kazi Fahmida Haque Shantanu[2], Shamiha Tabassum Teeya[1], Md. Motiar Rahman[3], A. K. M. Munzurul Hasan[4]*, Douglas P. Chivers[4], Tanveer A. Wani[5], Atekah Hazzaa Alshammari[6], Mahesh Rachamalla[4], Francisco Carlos da Silva Junior[7], Md. Munnaf Hossen[8]*

1 Department of Nutrition and Food Technology, Jashore University of Science and Technology, Jashore, Bangladesh, 2 Department of Botany, National University, Gazipur, Bangladesh, 3 Department of Chemistry, The State University of New York, Binghamton, New York, United States of America, 4 Department of Biology, University of Saskatchewan, Saskatoon, SK, Canada, 5 Department of Pharmaceutical Chemistry, King Saud University, Riyadh, Saudi Arabia, 6 Department of Biochemistry, King Saud University, Riyadh, Saudi Arabia, 7 Toxicology Centre, University of Saskatchewan, Saskatoon, SK, Canada, 8 School of Health and Biomedical Sciences, RMIT University, Bundoora, Victoria, Australia

* munzurul.hasan@usask.ca (AKMMH); munnafpstu@gmail.com (MMH)

## Abstract

The cytotoxic T-lymphocyte antigen-4 (CTLA4) is essential in controlling T cell activity within the immune system. Thus, uncovering the molecular dynamics of single nucleotide polymorphisms (SNPs) within the *CTLA4* gene is critical. We identified the non-synonymous SNPs (nsSNPs), examined their impact on protein stability, and identified the protein sequences associated with them in the human *CTLA4* gene. There were 3134 SNPs (rsIDs) in our study. Out of these, 186 missense variants (5.93%), 1491 intron variants (47.57%), and 91 synonymous variants (2.90%), while the remaining SNPs were unspecified. We utilized SIFT, PolyPhen-2, PROVEAN, and SNAP for identifying deleterious nsSNPs, and SNPs&GO, PhD SNP, and PANTHER for verifying risk nsSNPs in the *CTLA4* gene. Following SIFT analysis, six nsSNPs were identified as deleterious and reporting second and third nsSNPs as probably damaging and one as benign, respectively. From upstream analysis, rs138279736, rs201778935, rs369567630, and rs376038796 were found to be deleterious, probably damaging, and disease associated. ConSurf predicted conservation scores for four nsSNPs, and Project Hope suggested that all mutations could disrupt protein interactions. Furthermore, mCSM and DynaMut2 analyses indicated a decrease in ΔΔG stability for the mutants. GeneMANIA and STRING networks highlighted correlations with *CD86* and *CD80* genes. Finally, MD simulation revealed consistent fluctuation in RMSD and RMSF, consequently Rg, hydrogen bonds, and PCA in the mutant proteins compared with wild-type, which might alter the functional and structural stability of CTLA4 protein. The current comprehensive study shows how various nsSNPs in the *CTLA4* gene can modify the structural and functional characteristics of the protein, potentially influencing the pathogenesis of

**Data Availability Statement:** All relevant data are within the paper and its Supporting Information files.

**Funding:** This work was funded by the NSERC Discovery Grant awarded to D.P.C. and a grant (Project Number: RSP2024R357) from King Saud University, Riyadh, Saudi Arabia, for molecular dynamics simulations.

**Competing interests:** The authors have declared that no competing interests exist.

diseases in humans. Further, experimental studies are needed to analyze the effect of these nsSNPs on the susceptibility of pathological phenotype populations.

## 1. Introduction

Cytotoxic T-lymphocyte antigen 4 (CTLA4), also well-known as cluster of differentiation 152 (CD152), is a key immune regulatory protein which regulates T cell activation and immuno-logical homoeostasis [1, 2]. The *CTLA4* gene can inhibit T cell activation by attaching to cluster of differentiation 80 (CD80) and cluster of differentiation 86 (CD86) on antigen-presenting cells, avoiding excessive immunological activation and maintaining immune homoeostasis [3, 4]. This pathway is critical for autoimmunity prevention, inflammation control, and immu-nological tolerance regulation. The *CTLA4* gene, which comprises four exons and resides on chromosome 2q33, encodes the CTLA4 protein. The *CTLA4* gene exhibits a variety of genetic variants, including insertions, deletions, SNPs, and microsatellite repeats [5]. These variants may modify CTLA4 protein expression, structure, or function, altering immune control and perhaps having an impact on the onset of disease [6]. Association of *CTLA4* gene polymor-phisms have been found in Japanese patients with rheumatoid arthritis [7], South-Moroccan patients with type 1 diabetes (T1D) [8], Han-Chinese [9] and Asian [10] patients with breast cancer, as well as Iranian patients with gastric and colorectal cancers [11]. One study also found links between the *CTLA4* gene SNPs and autoimmune disorders [12]. The relationship between *CTLA4* gene SNPs and autoimmune thyroid illnesses like Graves' disease and Hashi-moto's thyroiditis highlights the potential contribution of genetic variants to the immunologi-cal dysregulation seen in these ailments. Furthermore, links between *CTLA4* gene polymorphisms and rheumatoid arthritis, T1D, systemic lupus erythematosus (SLE), and mul-tiple sclerosis have been found, pointing to a shared genetic vulnerability for autoimmune eti-ology [13–18].

*CTLA4* gene SNPs have been studied in relation to cancer in addition to autoimmune dis-eases [19]. For example, SNPs such as rs231775 (CT60), have been linked to elevated cancer risk and altered immunotherapy responses [20]. These results highlighted the importance of the *CTLA4* gene in tumor immunity and suggests the potential role of genetic variation in the progression and prognosis of cancer. Additionally, the relationship between CTLA4 polymor-phisms and infectious disorders has been studied. Certain *CTLA4* gene polymorphisms have been linked to hepatitis B and C infections, tuberculosis, and other diseases, suggesting that they may have an impact on how the body reacts to infectious agents [21]. Furthermore, it has been shown that CTLA4 polymorphisms affect the risk and severity graft-versus-host disease (GVHD) [22], a side effect of stem cell transplantation, albeit further research is needed to fully empathize the underlying mechanisms. The precise mechanisms behind how particular CTLA4 polymorphisms contribute to disease risk and development are still completely unidentified. For the complex interactions between the immune system and disease develop-ment to be fully understood, it is essential to comprehend the functional and structural impli-cations of these genetic variants.

Within the coding region, SNPs manifest in two forms: synonymous and non-synonymous SNPs (nsSNPs), with the latter type directly impacting protein sequences [23]. Those nsSNPs could change the stability, structure, and functions of the respective protein [24, 25]. The aim of this study was to use bioinformatics tools to predict nsSNPs in the *CTLA4* gene that may be deleterious. Subsequently, the *CTLA4* gene underwent comprehensive analysis to assess its pathogenicity.

## 2. Materials and methods

The entire process used in our investigation is outlined (Fig 1).

### 2.1. Retrieval of protein sequence and nsSNPs in CTLA4 gene

The protein sequence (FASTA format) and nsSNP of *CTLA4* gene were obtained using National Center for Biotechnology Information (NCBI) (https://www.ncbi.nlm.nih.gov/) and dbSNP database (https://www.ncbi.nlm.nih.gov/snp/), respectively (accessed November, 2024). Since missense mutations can prevent DNA-transcription, leading to alterations in the protein expression, nsSNPs were subsequently subjected to analysis using various bioinformatic methods [26].

### 2.2. Identification of deleterious nsSNPs in CTLA4 gene

SIFT web-server (https://sift.bii.a-star.edu.sg/) was employed to identify deleterious non-synonymous SNPs (nsSNPs) in *CTLA4* gene. SIFT assesses functional amino acids by presuming conservation of major amino acids, where changes at certain positions tend to be deleterious. SNPs with a probability score of ≤0.05 were classified as deleterious, while those with a score

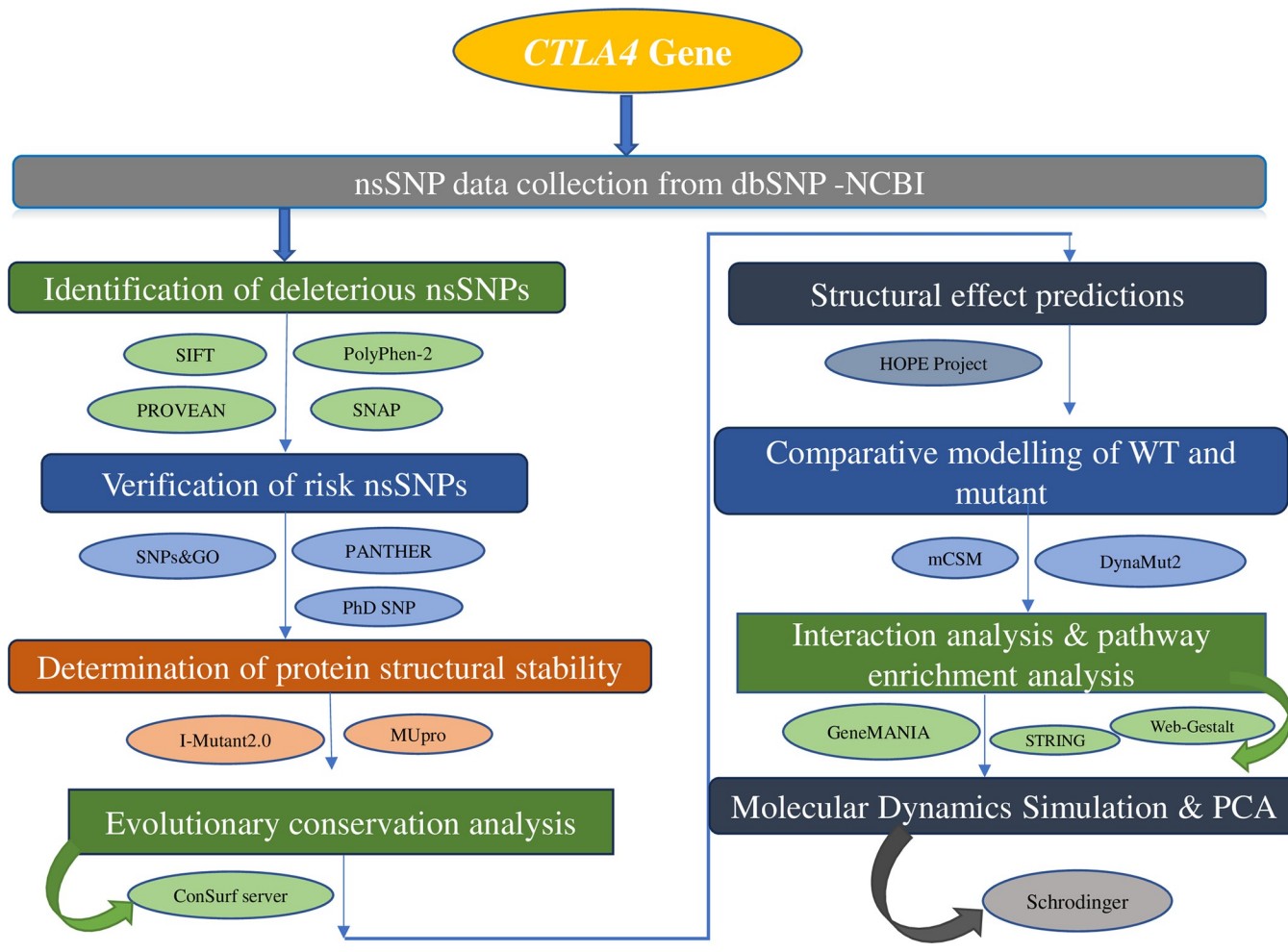

**Fig 1. The complete workflow employed in this study.**

>0.05 were considered tolerated. The analysis involved inputting single amino acid substitutions and rsIDs of each SNP in the human *CTLA4* gene [27].

PolyPhen-2 web-server (http://genetics.bwh.harvard.edu/pph2/) categorize and predict the functional effects of allele variations. It calculates the site-specific sequence conservation score of position-specific independent count (PSIC) and estimates the native and mutant variants differences. PolyPhen-2 classifies SNPs into three, 1) benign (0.00 to ≤0.45), 2) possibly damaging (>0.45 to ≤0.95), or 3) probably damaging (>0.95 to 1). The analysis entails inputting single amino acid changes and the FASTA sequence of the CTLA4 protein into PolyPhen-2 [28].

The PROVEAN web server (http://provean.jcvi.org) was employed to assess whether an amino acid alteration in protein sequences is harmful or neutral. It utilizes a delta alignment score derived from aligning homologous sequences with both the native and mutant protein sequences. Variants with a score of ≤ −2.5 are considered deleterious, while those with a score > −2.5 are deemed neutral. The input for PROVEAN includes the FASTA sequence of the CTLA4 protein and a list of SNPs [29].

SNAP, accessible at http://www.rostlab.org/services/SNAP, is a web-based tool utilized to assess the functional impact of amino acid substitutions in proteins using a neural network method. The tool rates each amino acid variation, identifying them as having either neutral or significant effects [30].

## 2.3. Verification of the risk nsSNPs

The SNPs&GO server (http://snps.biofold.org/snps-and-go/) was used to predict the effects of nsSNPs to evaluate the likelihood that each variant will be linked to a disease in humans. A prediction value of ≥0.5 indicates a variant's potential involvement in disease, whereas a value <0.5 suggests a neutral impact. Additionally, this service combines its predictions with results from the PhD-SNP and PANTHER for comprehensive analysis [31].

## 2.4. Determination of protein structural stability using I-Mutant2.0 and MUpro server

I-mutant2.0 (http://folding.biofold.org/i-mutant/i-mutant2.0.html) was utilized to predict the potential effects of nsSNPs on protein stability. Following this analysis, Delta Delta G (ΔΔG) values were predicted. Protein stability rises with a ΔΔG score ≥0 kcal/mol and falls with a ΔΔG score <0 kcal/mol. The protein's FASTA sequence, the location of the amino acid substitution, and the variant residue were entered for analysis [32].

The MUpro server (http://mupro.proteomics.ics.uci.edu/) is used to forecast how amino acid substitutions might lower or raise the stability of a respective protein. This server employs both neural networks and support vector machines (SVM) for its predictions. A confidence score <0 indicates a decrease in protein stability, while a score ≥0 suggests an increase [33].

## 2.5. Evolutionary conservation analysis by ConSurf

The ConSurf web server (http://consurf.tau.ac.il/) was employed to trace the evolutionary conservation of amino acid residues (wild type) and to pinpoint nsSNPs at particular positions. By generating a phylogenetic tree from the phylogenetic relationships among homologous sequences, this server calculates the evolutionary trajectories of amino acid sites within a protein [34].

## 2.6. Prediction of structural effects of CTLA4 mutants

The Project Hope web server (http://www.cmbi.ru.nl/hope/) was used to assess the structural and functional impact of point mutations. The HOPE server provides a 3D visualization of the

mutated proteins, incorporating predictions from UniProt and DAS servers. Inputs included the protein sequence, wild type (WT), and mutated amino acids, with the results presented in text, graphical, and animated formats [35].

## 2.7. Comparative modelling of wild-type CTLA4 protein and their mutant structures

The mCSM (http://structure.bioc.cam.ac.uk/mcsm) is a machine learning method to predict the effects of missense mutations based on the structural characteristics of the corresponding protein. Here is the mCSM determination of the effects of missense variants on CTLA4 stability. The tools predict changes in protein folding free energy (ΔΔG) as a consequence of missense mutations and are classified into two categories (stabilizing and destabilizing) based on the ΔΔG [36]. The DynaMut2 web-server (https://biosig.lab.uq.edu.au/dynamut2/) was utilize to calculate the impact on the flexibility and stability of proteins. Using the normal mode analysis (NMA) approach, DynaMut2 was utilized to determine how the mutation affected protein stability and dynamics. The predicted Gibbs free energy (G) values of mutants <0 was categorized as destabilizing, and those >0 as stabilizing. Both single mutations and multiple mutations can be predicted by DynaMut2, as it utilizes a single mutation prediction feature for the corresponding protein. We inputted a list of mutations and the wild-type structure in PDB format [37].

## 2.8. Gene-gene, protein-protein interactions and pathway enrichment analysis

GeneMANIA (http://www.genemania.org) is a web-based tool that sifts through an extensive database to discover supplementary genes associated with a provided set of input genes. Examples of associated data include co-expression, pathways, colocalization, protein domain similarity, and protein-protein interactions. We searched GeneMANIA using *CTLA4* as an input gene for *Homo sapiens* [38]. Protein-protein interactions are crucial for determining how proteins function and play specific roles in disease processes as proteins are a crucial component of molecular pathways. The Search Tool for the Retrieval of Interacting Genes (STRING) (https://string-db.org/) is a biological database and online tool used for interpreting the protein-protein interactions. A search conducted using the terms "*CTLA4* gene" and "Homo sapiens" as input genes, using the highest confidence score of 0.999, we identified ten interacting proteins [39]. The publicly available Web-Gestalt (WEB-based Gene SeT AnaLysis Toolkit) (http://www.webgestalt.org/) was used to analyze the pathways enrichment analysis of the associated gene [40].

## 2.9. Molecular dynamics simulation analysis

A 100 ns MD simulation analysis were performed under the Linux framework in Schrodinger 2020–3 with "Desmond v6.3 Program" to assess the structural stability of the corresponding protein [41]. The three-site transferrable intermolecular potential (TIP3P) water model was used to analyze the MD simulation [42]. An orthorhombic box shape was used to maintain a specified volume and Na+ and Cl- were added to neutralize the whole system with a salt concentration of 0.15 M. An OPLS3e force field was applied [43]. Further the protein structure system minimized using a natural time and pressure (NPT) ensemble at a constant pressure of 101325 Pascal's and a temperature of 300 K. The protein stability and dynamic properties were assessed using the RMSD (root means square deviation), RMSF (root means square fluctuation), Rg (radius of gyration), and hydrogen bonds. Principal Component Analysis (PCA) has

been widely used to study the slow and functional movements of biomolecules [44]. Prior to performing PCA, the correlation matrix C needed to be calculated. These are the definitions of the parts $C_{ij}$ in the matrix, $C_{ij} = \langle (x_i - \langle x_i \rangle)(x_j - \langle x_j \rangle) \rangle$ where $x_i$ and $x_j$ are instant coordinates of atoms and $\langle x_i \rangle$ and $\langle x_j \rangle$ are the average coordinate over the group. Principal Component Analysis (PCA) has been widely used to study the slow and functional movements of biomolecules [44]. Prior to performing PCA, the correlation matrix C needed to be calculated.

## 3. Results

### 3.1. Retrieval of protein sequence and nsSNPs in CTLA4 gene

Following dbSNP database, we have found a total of 3134 SNPs. Following these, about 186 missense variants (5.93%), 1491 intron variants (47.57%), and 91 synonymous variants (2.90%) were computed (Fig 2A). The protein sequence (UniProt ID: P16410) of *CTLA4* gene has been retrieved from NCBI. In this study, missense variants were subjected to further computational analysis to explore the most deleterious nsSNPs.

### 3.2. Identification of deleterious nsSNPs in CTLA4 gene

Following the analysis of 186 nsSNPs using SIFT, 18 nsSNPs were identified, with 12 being eliminated as they were deemed tolerated by SIFT. Among the remaining 6 nsSNPs, those with a tolerance index of $\leq 0.05$ were predicted as deleterious. Polyphen-2 produced two

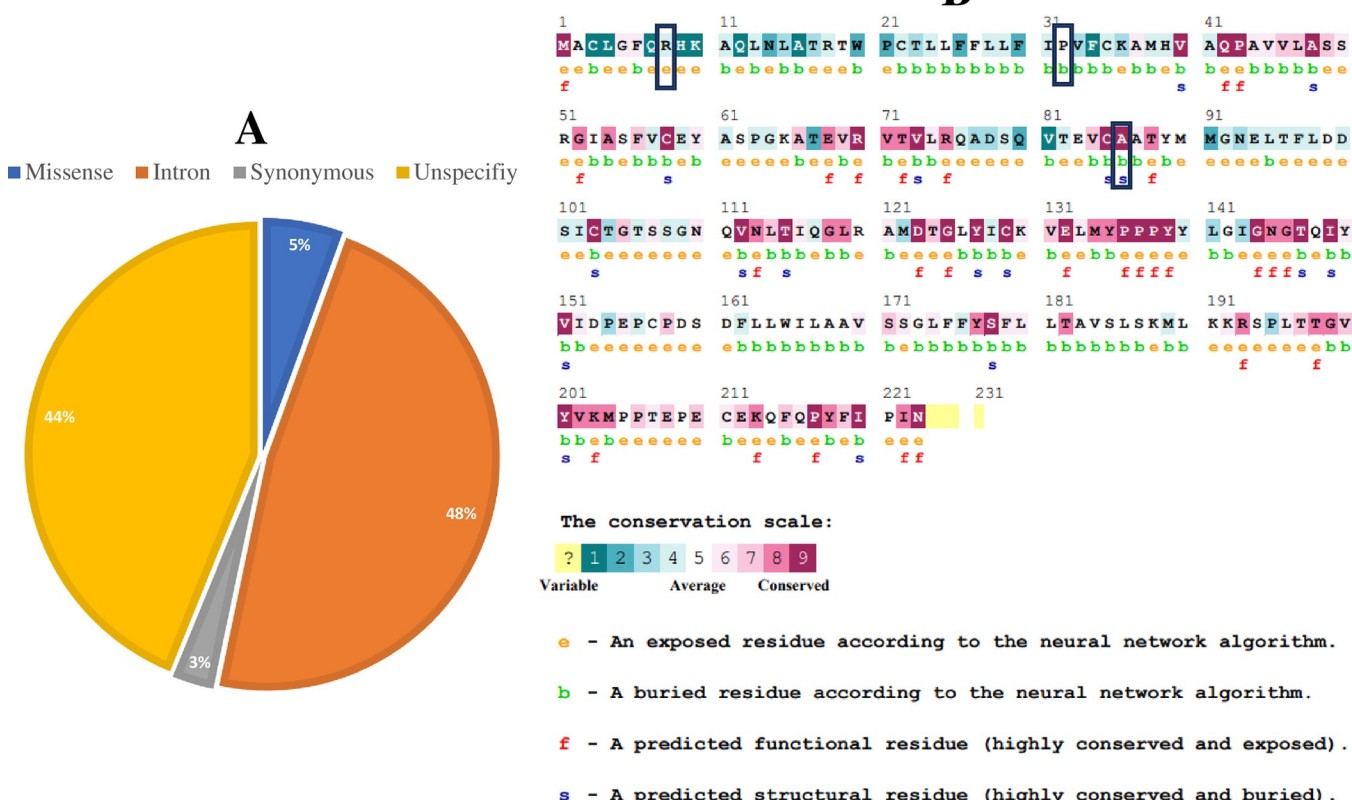

**Fig 2. Retrieved SNPs of *CTLA4* gene and evolutionary conservation analysis.** (A) Pie chart representing total SNPs of *CTLA4* gene. (B) Prediction of evolutionary conserved amino acid residues by ConSurf server. Conservation score is represented as the color-coding bars.

scores, HumDiv and HumVar, for the 6 nsSNPs analyzed. HumDiv identified 2 nsSNPs as probably damaging with high confidence, 1 as possibly damaging, and 1 as benign. In contrast, HumVar classified 3 nsSNPs as probably damaging with high confidence and 1 as benign. Subsequent analysis with the PROVEAN program revealed that 3 out of the 4 nsSNPs in the *CTLA4* gene were deleterious. Additionally, the SNAP program was employed to evaluate the impact of 4 nsSNPs on the protein sequence, identifying 3 as having non-neutral effects.

### 3.3. Verification of the risk nsSNPs

We employed the SNPs&GO, PhD-SNP, and PANTHER programs to forecast disease-related nsSNPs. Out of the 4 nsSNPs examined, SNPs&GO identified 4, PhD-SNP recognized 3, and PANTHER detected 3 nsSNPs linked to disease (Table 1). Upon comparing the outcomes from analyses of the 4 nsSNPs using computational tools including SIFT, Polyphen-2, PROVEAN, SNAP, SNPs&GO, PhD-SNP, and PANTHER, we observed a strong concordance among the 7 programs regarding nsSNPs rs138279736, rs201778935, rs369567630 and rs376038796.

### 3.4. Determination of protein structural stability I-Mutant2.0 and MUpro server

The structural stability of the protein was assessed using both I-Mutant2.0 and the MUpro server. These servers predict the structural impact of four selected nsSNPs by analyzing the change in free energy ($\Delta\Delta G$) and reliability index (RI) following mutation. I-Mutant2.0 predicted the stability changes caused by the second nsSNP, while MUpro predicted those caused by the third nsSNP in the CTLA4 protein (Table 2).

### 3.5. Evolutionary conservation analysis by ConSurf

Using the ConSurf tool, a color-coded conservation score was generated for the CTLA4 protein, highlighting highly conserved functional areas. Among the four most harmful SNPs, ConSurf predicted two amino acids (R8Q and R8W) with a conservation score of 4, which are

**Table 1. Screening of deleterious single nucleotide polymorphism (SNP) predicted by SIFT, Polyphen-2, PROVEAN, SNAP, SNPs&GO, PhD-SNP, and Panther.**

| SNP Id | Allele | Variant | SIFT | Polyphen-2 | | Provean | SNAP | SNPs&GO | PhD SNP | Panther |
|---|---|---|---|---|---|---|---|---|---|---|
| | | | | HumDiv | HumVar | | | | | |
| rs138279736 | G/A | R8Q | D | poss | Pro | D | E | N | Dis | N |
| rs201778935 | C/T | R8W | D | pro | Pro | D | E | N | Dis | Dis |
| rs369567630 | C/T | P32S | D | benign | benign | D | N | Dis | Dis | Dis |
| rs376038796 | C/T | A86V | D | pro | Pro | N | E | Dis | N | Dis |

D: Deleterious, E: Effect N: neutral, Dis: Disease, Pro: Probably damaging (high confident), Poss: Possibly damaging (low confident).

**Table 2. Characterization of the effect of deleterious SNPs on protein stability by I-mutant2.0 and MUpro.**

| SNP ID | Variant | I-Mutant2.0 | | MUpro | |
|---|---|---|---|---|---|
| | | Stability | DDG (kcal/mol) | Prediction | Confidence score |
| rs138279736 | R8Q | Increase | 0.41 | Decrease | -1.0 |
| rs201778935 | R8W | Increase | 0.12 | Decrease | -1.0 |
| rs369567630 | P32S | Decrease | -0.82 | Decrease | -0.84 |
| rs376038796 | A86V | Decrease | -0.20 | Increase | 0.44 |

**Table 3. The effect mutation on protein using Project Hope prediction.**

| Rs Id | Variant | Wild Type | Mutant Type |
|---|---|---|---|
| rs138279736 | R8Q | Arginine | Glutamine |
| rs201778935 | R8W | Arginine | Tryptophan |
| rs369567630 | P32S | Proline | Serine |
| rs376038796 | A86V | Alanine | Valine |

exposed residue and rest two (P32S and A86V) with a score of 5 and 9; buried and structural residue respectively (Fig 2B). Highly conserved regions typically play a critical role in the biological function of the respected protein.

### 3.6. Prediction of structural effects of CTLA4 mutants

The Project Hope server assesses how mutations affect the structural characteristics of a protein, considering factors (size, charge, hydrophobicity, and spatial structure) in comparison to the wild type (WT). Maintaining protein stability depends critically on the size of amino acid residues. Alterations in residue size can impair folding, structural integrity, or functional interactions, resulting in protein failure. Among the four anticipated substitutions, the mutant residues R8Q and R8W were smaller than the wild-type residues, while the mutants P32S and A86V were larger. Consequently, mutating these residues may disrupt protein interactions (Table 3).

### 3.7. Comparative modelling of WT CTLA4 protein and their mutant structures

We used mCSM to assess the impacts of missense variants on CTLA4 stability. The mCSM relies on graph-based signatures to predict the impacts of missense variants on protein stability. Here, all four variants destabilized the protein (Table 4). DynaMut2 was used to calculate the general dynamic traits of the highest deleterious nsSNPs including R8Q, R8W, P32S and A86V mutants (Table 5). DynaMut2 interprets the predictions for $\Delta\Delta G$ stability value among the WT and mutant CTLA4 protein. All mutants showed a decrease in the $\Delta\Delta G$ stability value compared to WT and were found to be responsible for destabilizing the protein. The free energy ($\Delta\Delta G$) stability values for R8Q, R8W, P32S and A86V mutants were found -0.51, -0.34, -0.96 and -0.94 kcal/mol respectively.

### 3.8. Gene-gene, protein-protein interactions and pathway enrichment analysis

Protein interactions are fundamental in determining the resilience and flexibility of biological networks. Disruptions in these relationships may result in network malfunction, significantly impacting cellular and organismal health. The interaction network between genes associated with the *CTLA4* gene demonstrated several correlations. Gene-gene interaction networks and

**Table 4. Alterations in stability of protein and interaction upon nsSNPs.**

| SL | RsID | Variants | RSA (%) | DDG Value | Prediction |
|---|---|---|---|---|---|
| 01 | rs138279736 | R8Q | 100 | -0.007 | Destabilizing |
| 02 | rs201778935 | R8W | 100 | -0.369 | Destabilizing |
| 03 | rs369567630 | P32S | 74 | -0.806 | Destabilizing |
| 04 | rs376038796 | A86V | 19.2 | -0.498 | Destabilizing |

**Table 5. Alterations in structural stability and molecular interactions due to nsSNPs.**

| SL | RsID | Variant | ΔΔG | ΔΔG$^{Stability}$ |
|----|------|---------|------|-------------------|
| 01 | rs138279736 | R8Q | -0.51 kcal/mol | Destabilising |
| 02 | rs201778935 | R8W | -0.34 kcal/mol | Destabilising |
| 03 | rs369567630 | P32S | -0.96 kcal/mol | Destabilising |
| 04 | rs376038796 | A86V | -0.94 kcal/mol | Destabilising |

functional analyses highlighted gene sets enriched within the CTLA4 target network. Various interactions were represented by different colors of network edges, including Physical Interactions, Co-expression, Co-localization, Predicted, Genetic Interactions, Pathway, and Shared protein domains. GeneMANIA constructed a composite gene-gene functional interaction network. It enhances our comprehension of the molecular mechanisms that elucidate observable phenotypes. The study identified an association between the *CTLA4* gene and 20 other genes, with *CD86*, *CD80*, and *adaptor related protein complex 2 subunit mu 1 (AP2M1)* (Fig 3A and Table 6).

Proteins collaborate and interact with others to facilitate cell signaling and various cellular functions. Consequently, variations in amino acids within a protein can impact other proteins within the network. The STRING tool predicted the functional partners of *CTLA4* gene. CTLA4, also known as Cytotoxic T-lymphocyte protein 4, serves as a significant negative regulator of T-cell responses. It exhibits notably stronger affinity towards its natural B7 family ligands, *CD80* and *CD86*, compared to the stimulatory coreceptor *cluster of differentiation 28 (CD28)*. LTF emerged as the primary interacting partner of the T-lymphocyte activation antigen *CD80* (Fig 3B).

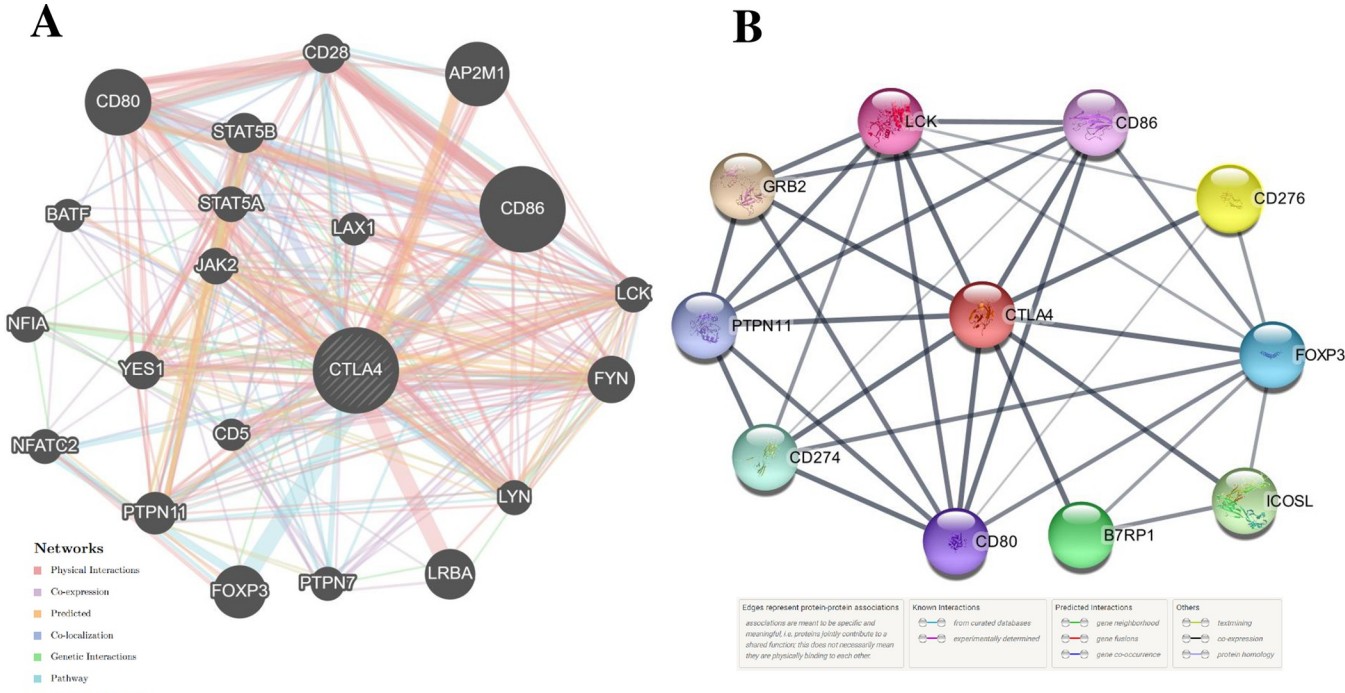

**Fig 3. Gene-gene and protein-protein interactions analysis.** (A) GeneMANIA gene-gene interaction for *CTLA4* Gene Different colors of the network edge indicate: co-expression, website prediction, pathway, physical interactions and co-localization. (B) CTLA4 interacts with a total of 10 different proteins. Colored nodes: query proteins and first shell of interactors. Empty nodes: proteins of unknown 3D structure. Filled nodes: 3D structure is known/predicted. Edges represent protein-protein associations.

**Table 6. Gene functionally linked to *CTLA4* identified using GeneMANIA.**

| Gene | Description |
|------|-------------|
| CTLA4 | cytotoxic T-lymphocyte associated protein 4 |
| CD86 | CD86 molecule |
| CD80 | CD80 molecule |
| AP2M1 | adaptor related protein complex 2 subunit mu 1 |
| FOXP3 | forkhead box P3 |
| LRBA | LPS responsive beige-like anchor protein |
| FYN | FYN proto-oncogene, Src family tyrosine kinase |
| PTPN11 | protein tyrosine phosphatase non-receptor type 11 |
| STAT5B | signal transducer and activator of transcription 5B |
| CD28 | CD28 molecule |
| YES1 | YES proto-oncogene 1, Src family tyrosine kinase |
| JAK2 | Janus kinase 2 |
| NFIA | nuclear factor I A |
| STAT5A | signal transducer and activator of transcription 5A |
| LCK | LCK proto-oncogene, Src family tyrosine kinase |
| LYN | LYN proto-oncogene, Src family tyrosine kinase |
| NFATC2 | nuclear factor of activated T cells 2 |
| PTPN7 | protein tyrosine phosphatase non-receptor type 7 |
| BATF | basic leucine zipper ATF-like transcription factor |
| LAX1 | lymphocyte transmembrane adaptor 1 |
| CD5 | CD5 molecule |

The KEGG pathway enrichment analysis, based on the highest enrichment ratio and FDR value revealed that those gene regulate several metabolic pathways (S1 Table). According to this analysis, the most enriched pathways were PD-L1 expression and PD-1 checkpoint pathway in cancer, T cell receptor signaling pathway, Th17 cell differentiation, Cell adhesion molecules and Autoimmune thyroid disease (S1 Table).

## 3.9. Molecular dynamics simulation analysis

The structural stability of native and four mutant proteins were calculated (RMSD and RMSF) and compared to evaluate structural and functional change due to the mutation of the corresponding protein. Here, 100 ns MD simulation analysis were conducted to observe this deviation in an artificial environment. Mutants R8Q, R8W, P32S and A86V showed considerable fluctuations compare with the native protein CTLA4. The RMSD values for the native protein were observed ranging from 0.325 nm to 4.038 nm while 0.398 nm to 3.980 nm for the R8Q mutant, 0.364 nm to 3.628 nm for the R8W mutant, 0.323 nm to 3.968 nm for the P32S mutant, 0.317 nm to 4.436 nm for the A86V mutant. The average RMSD values of native CTLA4, and mutant R8Q, R8W, P32S and A86V are 3.103 nm, 2.987 nm, 2.759 nm, 3.164 nm, and 3.680 nm, respectively (Fig 4). Overall, the RMSD plot showed that the native protein was stable along the simulation run time and also demonstrated that the mutant protein has a major impact on the structural confirmation of the CTLA4 protein. The RMSF of protein residue was counted to observe how these changes impact the local flexibility. In addition, RMSF analysis demonstrated considerable fluctuations between native and mutant structures throughout the simulation run. The average RMSF values are 1.102 nm, 0.907 nm, 0.886 nm, 0.87 nm, 0.891 nm, and 1.060 nm correspondingly (Fig 5). The Rg measures how the atoms are distributed around the axis of a protein. It is an important indicator for predicting the

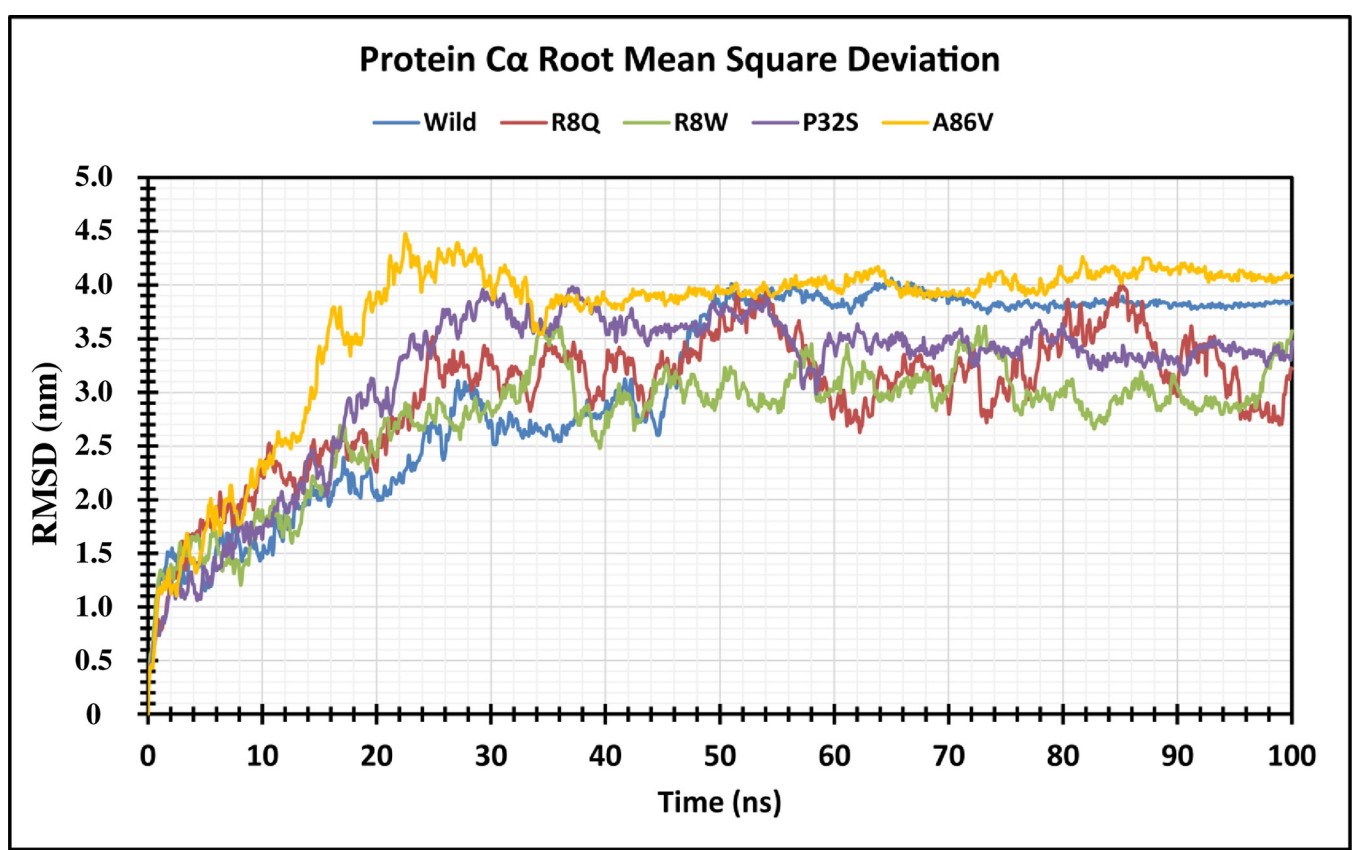

**Fig 4. RMSF analysis shows Cα atom fluctuations over 100 ns simulation.** The color scheme is as follows: native (blue), R8Q mutant (red), R8W mutant (green), P32S mutant (violet), and A86V mutant (orange). RMSF analysis shows Cα atom fluctuations over 100 ns simulation.

structural activity of macromolecules and for assessing changes in the compactness of the protein structure. The native protein and four mutants R8Q, R8W, P32S and A86V revealed Rg ranges of 2.189 nm to 5.025 nm, 2.651 nm to 4.877 nm, 2.246 nm to 5.249 nm, 2.181 nm to 5.060 nm, and 1.982 nm to 5.189 nm, respectively. Average fluctuation of these mutants was 3.108 nm, 3.297 nm, 3.406 nm, 2.856 nm, and 2.583 nm, respectively (Fig 6). The mutated protein structures were unstable in 100 ns simulations with greater fluctuation differences from lowest to highest, suggesting that the mutated proteins significantly alter the conformation of the respective protein's active site. The number of hydrogen bonds can help to characterize a protein. Therefore, hydrogen bond numbers were calculated from initial to final times during the 100 ns simulation run to observe each hydrogen bond. All the proteins formed several hydrogen bonds ranging between 128 to 184 occur simultaneously until 100 ns simulation time (Fig 7 and S2 Table). These results indicated that the native protein has a higher specificity while the mutated protein has a higher fluctuating range of hydrogen bonds denoted the structural instability. The PCA plots provide insights into the genetic variance and clustering patterns for four mutations compared to the wild-type. Each mutant SNP variant exhibits unique clustering patterns compared to the wild-type. The difference patterns in the PCA plots show that each mutation (R8W, R8Q, P32S, and A86V) might change structure or function in a way that is different from the wild-type (Fig 8).

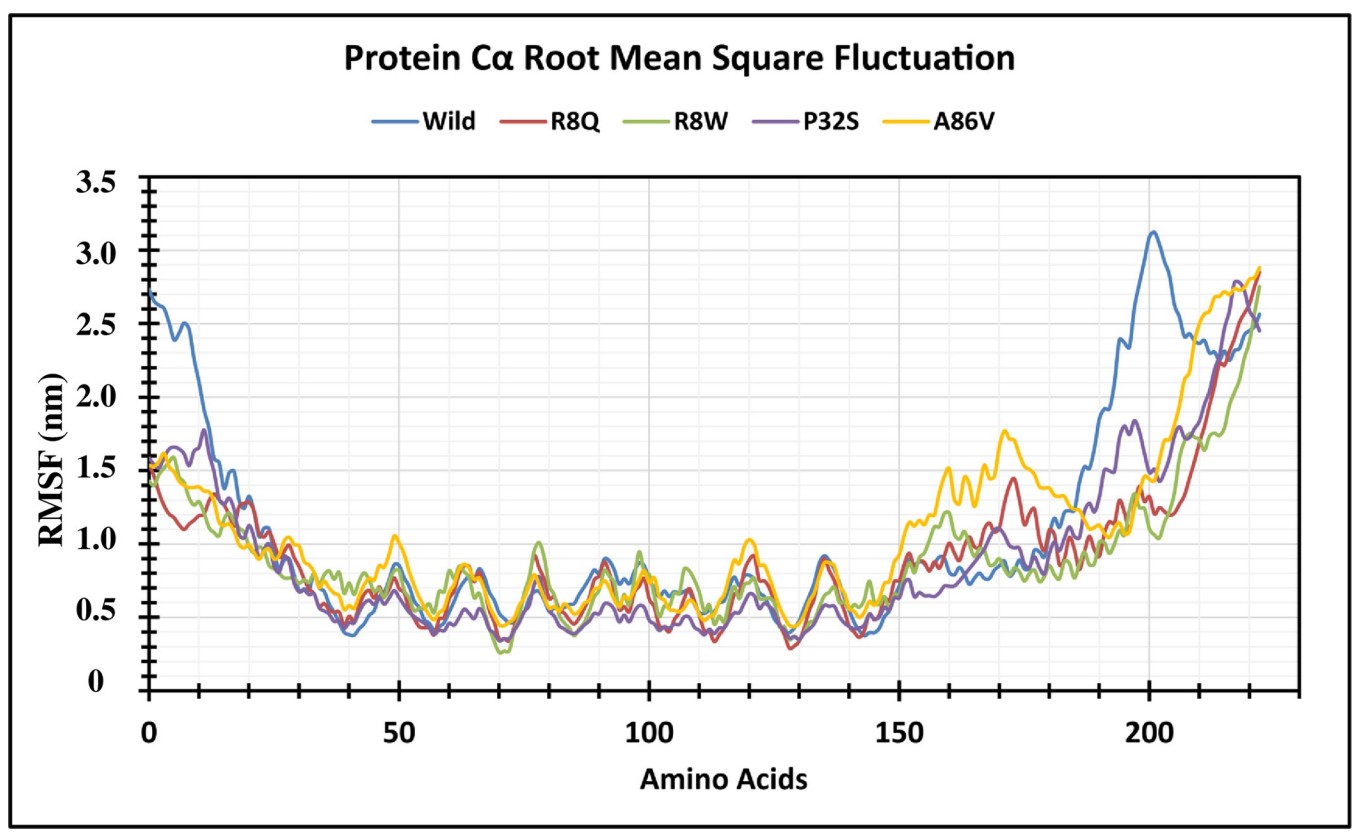

**Fig 5. Root mean square fluctuation (RMSF) analysis reveals variations in Cα atom dynamics over 100 ns.** The color scheme is as follows: native (blue), R8Q mutant (red), R8W mutant (green), P32S mutant (violet), and A86V mutant (orange).

## 4. Discussion

The human Cytotoxic T-lymphocyte antigen 4 (hg-CTLA4) serves as a crucial negative regula-
tor in the immune system, particularly for regulatory T cells (Tregs), which play a role in sup-
pressing T cell proliferation and differentiation. Constitutively expressed by Tregs, hg-CTLA4
is also induced in activated conventional T cells [1, 2, 45, 46]. Mutations in the *CTLA4* gene
have been associated with a range of clinical manifestations, encompassing different autoim-
mune conditions affecting specific organs, hypo-gammaglobulinemia, recurrent infections,
and cancer [47]. Missense or non-synonymous mutations contribute to protein destabilization
and can influence susceptibility to diseases as well as responses to drug treatments [48, 49].
Numerous SNPs within the *CTLA4* gene have been identified and documented in the dbSNP
database. Consequently, we conducted systematic and comprehensive bioinformatic analyses
to uncover the functional and structural impact of nsSNPs within the human *CTLA4* gene.
Our aim was to understand how these mutations affect the gene's functionality and its role in
disease pathogenesis. Through in silico structural and functional analyses, we identified poten-
tial nsSNPs within the *CTLA4* gene.

Seven web servers, namely SIFT, PolyPhen-2, PROVEAN, SNAP, SNPs&GO, PhD-SNP,
and PANTHER, were utilized to identify the most detrimental nsSNPs. By comparing the
scores from all servers, we determined that four nsSNPs R8Q, R8W, P32S, and A86V were
considered deleterious, likely causing damage, affecting protein function, and associated with
diseases [50]. These nsSNPs were then selected for further analysis using additional *in silico*
tools. Stability is a vital property of corresponding protein that affects the function, activity,

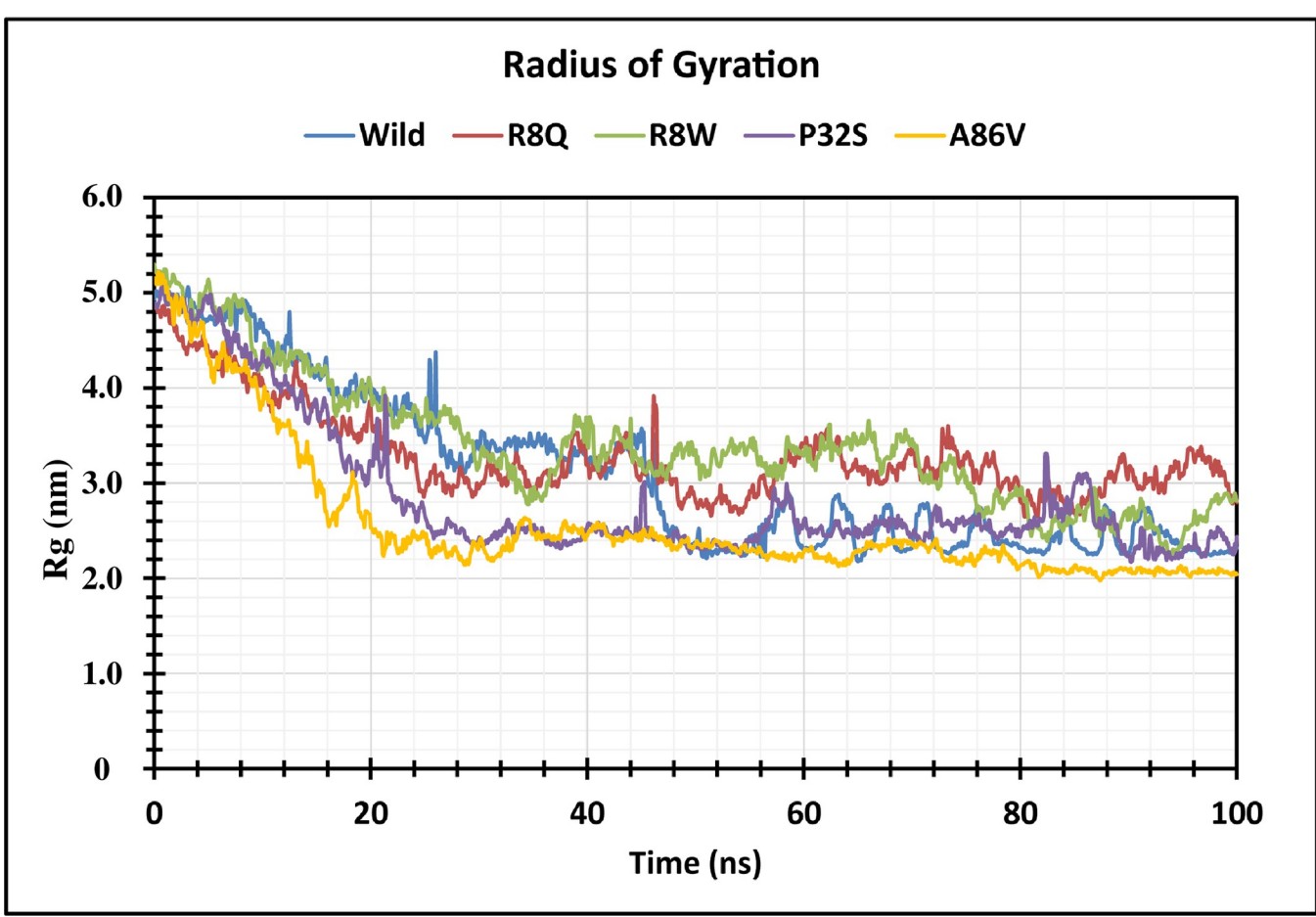

**Fig 6. Radius of gyration (Rg) analysis shows structural stability differences in wild-type and mutant proteins.** It is represented as a time-dependent change during the simulation. The color scheme is as follows: native (blue), R8Q mutant (red), R8W mutant (green), P32S mutant (violet), and A86V mutant (orange).

and regulation of protein. Stability changes occurred due to mutation of proteins which are involved in diseases [51]. Using the I-Mutant2.0 and MUpro programs, protein stability changes were observed. In P32S and A86V nsSNPs, and R8Q, R8W, and P32S nsSNPs were decreased protein stability using I-Mutant2.0 and MUpro programs respectively. Subsequently, the ConSurf tool generated a color-coded conservation score for the CTLA4 protein, highlighting regions of high conservation. Within the four most harmful nsSNPs, ConSurf identified two amino acids with a conservation score of 4, one with a score of 5, and one with a score of 9. These conserved residues typically play critical roles in biological function. When a highly conserved residue undergoes mutation, it tends to have more detrimental effects compared to a less conserved one [24].

Once more, HOPE offers 3D structural depictions of both mutated and wild-type (WT) proteins. Predicted substitutions from the HOPE server indicated that mutant residues R8Q and R8W were smaller than the wild residues, while P32S and A86V mutants were larger. This disparity could result in an unoccupied area within the protein's core, potentially leading to the loss of hydrophobic interactions. Such vacancies might affect the protein's function, properties, or reactivity [52]. Those variants were analyzed following mCSM, where $\Delta\Delta G < -0$ kcal/mol were considered to reduce CTLA4 stability (destabilizing). Here, all four variants

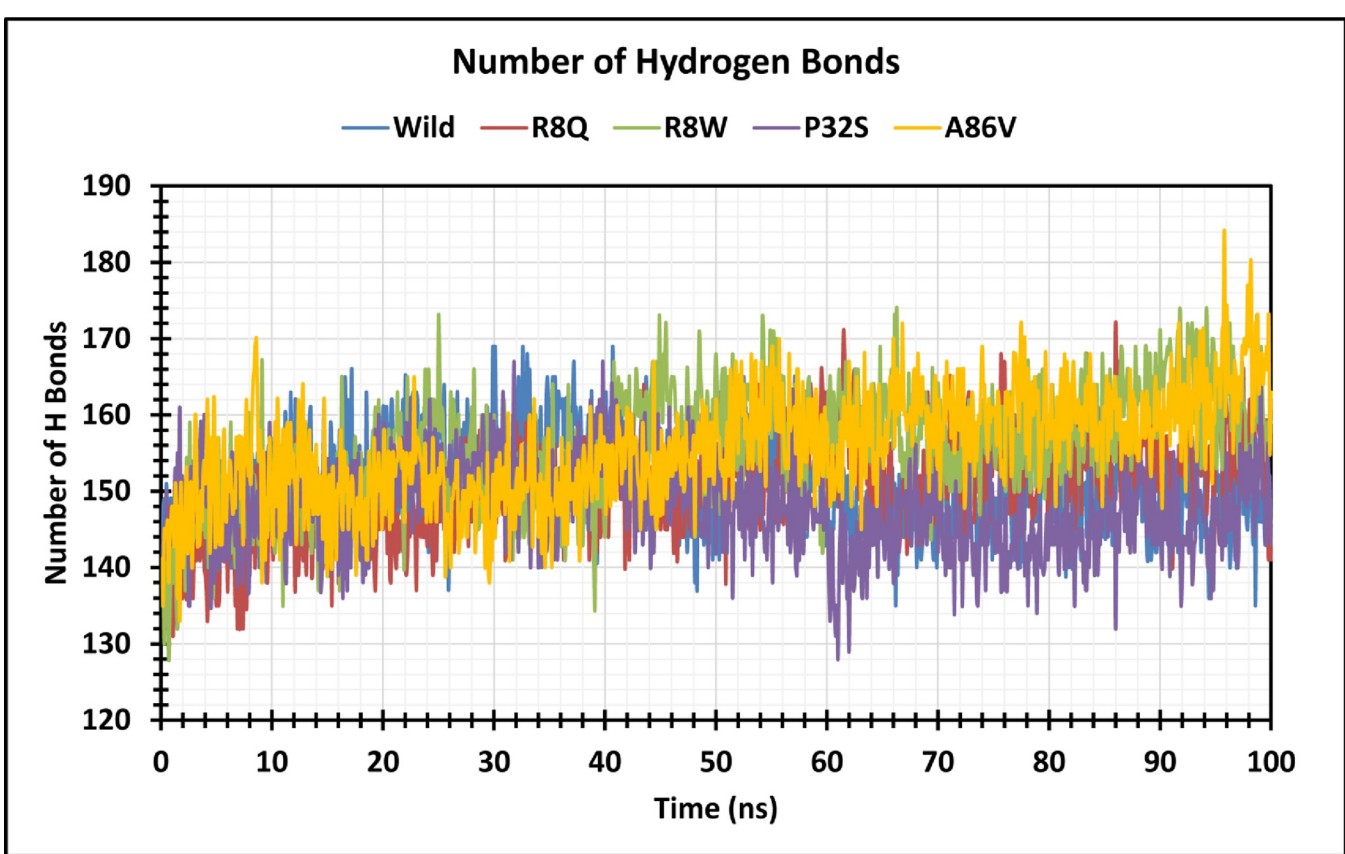

**Fig 7. Analysis of hydrogen bonds in wild type and mutant proteins over 100 ns simulation conducted.** The color scheme is as follows: native (blue), R8Q mutant (red), R8W mutant (green), P32S mutant (violet), and A86V mutant (orange).

resulted in reduced protein stability (Table 4). The WT and mutant CTLA4 protein ΔΔG stability value for each protein structure was predicted using DynaMut2. The free energy (ΔΔG) stability values for R8Q, R8W, P32S and A86V mutants were found -0.51, -0.34, -0.96 and -0.94 kcal/mol respectively (Table 5). All mutants showed a decrease in the ΔΔG stability value compared to WT and were found to be responsible for destabilizing the protein. This difference in the free energy landscape explains how the mutation influences the protein's stability [37].

GeneMANIA analysis showed that CTLA4 interacts significantly with 20 different proteins, which are particularly crucial in neurogenesis. Among these, CD86, CD80, and AP2M1 exhibit particularly significant interactions with CTLA4. CD86 is involved in regulating B-cell IgG1 production levels and signaling for self-regulation and cell-cell association or disassociation. Similarly, CD80, also known as the CD80 molecule, plays a role in T-cell activation and in regulating the activity of both normal and malignant B cells [53, 54]. Together with CD80, CD86 delivers costimulatory signals crucial for T cell activation and survival. AP2M1 facilitates autophagy-induced degradation of CLDN2 through endocytosis and interaction with LC3, consequently reducing intestinal epithelial tight junction permeability. The String tool was utilized to anticipate the close interactor proteins of CTLA4 (Fig 3B). Proteins operate in a concerted manner and interact with others to execute cell signaling and various cellular functions. CTLA4 exhibits notably stronger affinity towards its natural B7 family ligands, CD80 and CD86, compared to the stimulatory coreceptor CD28. LTF emerged as the primary interacting

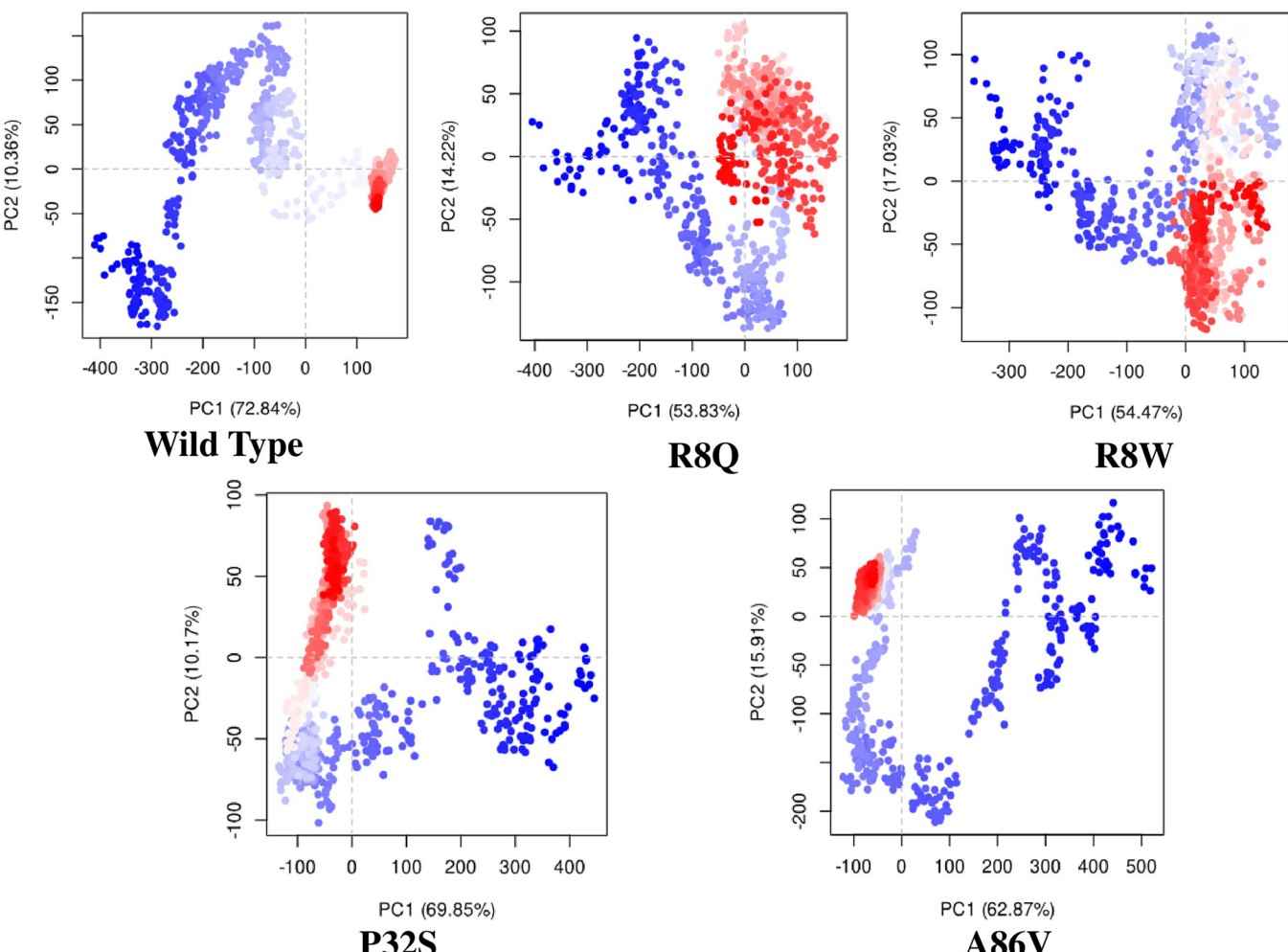

**Fig 8. Principal component analysis (PCA) of wild type and mutant proteins for molecular dynamics simulation trajectories.**

partner of the T-lymphocyte activation antigen CD80. The pathway enrichment analysis revealed that genes interacting with the *CTLA4* gene were associated with disease and therefore, might play a significant regulatory network in the progression of disease pathogenesis [55]. In order to investigate the impact of alterations on the protein structure, 100 ns MD simulations were utilized for the above-mentioned variations. This analysis was conducted to observe this deviation in an artificial environment. Mutants R8Q, R8W, P32S and A86V showed considerable fluctuations compare with the native protein CTLA4. The RMSD of protein residue was counted to observe how these changes impact the local flexibility (Fig 4). In addition, RMSF analysis demonstrated considerable fluctuations between native and mutant structures throughout the simulation run (Fig 5). The protein structure is stabilized when the RMSD and RMSF values of a protein are within 0.1–0.3 nm [56]. The Rg plot displays against time for all mutant proteins as well as the wild type protein. It was discovered that the average Rg of the wild type protein was 3.108 nm, whereas the mutant was 3.297 nm, 3.406 nm, 2.856 nm, and 2.583 nm, respectively (Fig 6). Similarly, the increased number of hydrogen bonds causes the structural unsteadiness of the corresponding protein (Fig 7). The PCA analysis provide insights into the genetic variance and clustering patterns. The difference patterns in the

PCA plots show that each mutation (R8W, R8Q, P32S, and A86V) might change structure or function in a way that is different from the wild-type. The PCA study reveals genetic differences between wild-type and each SNP mutation add discrete genetic diversity (Fig 8). These can direct next research on the possible consequences of any mutation on protein stability, function, or phenotype.

In Chinese Han population and Egyptian, rs231775 of *CTLA4* gene has found involvement in hepatocellular carcinoma and a direct association with cancer [57, 58]. In Pakistan, these rs231775 are also associated with HCC [59]. In *CTLA4* gene, rs221775 the variant is associated with multiple sclerosis susceptibility [60]. Similarly, rs11571317 and rs3087243 variants play role in breast cancer progression [61]. Genetic variants rs3087243 and rs231775 have an association with Graves' disease in Chinese Han population [62]. Interestingly, three major high-risk SNPs, rs1553657429, rs1559591863, rs778534474 also have been found within CTLA4 gene [63]. Also, association of CTLA-4 gene polymorphisms were found in Japanese patients with rheumatoid arthritis [7], South Moroccan with type 1 diabetes [64], Chinese Han [9] and Asian populations [10], and Iranian patients with gastric and colorectal cancers [11].

The non-synonymous SNPs (nsSNPs) R8Q, R8W, P32S, and A86V (rs138279736, rs201778935, rs369567630, and rs376038796) have not yet been examined, underscoring the need for research that accounts for the ethnic variability in disease susceptibility across diverse populations. This study has identified SNPs within the *CTLA4* gene that may potentially interfere with ligand-receptor interactions, suggesting a significant risk. Nevertheless, additional *in-vitro* and *in-vivo* experimentation, as well as genetic association studies in human population are needed to validate the impact of these nsSNPs on the pathological phenotypes and drive future progress about genetic polymorphism research.

## 5. Conclusion

Currently, *in silico* methodologies are gaining prominence as a crucial strategy for identifying SNPs associated with diseases. In the present work, a thorough investigation of the *CTLA4* gene was conducted using various computational tools to explore how non-synonymous SNPs (nsSNPs) impact the protein's structure and functionality. The study identified 3134 SNPs within the *CTLA4* gene, encompassing 186 missense variants (5.93%), 1,491 intron variants (47.57%), and 91 synonymous variants (2.90%). Four specific nsSNPs within the *CTLA4* gene rs138279736, rs201778935, rs369567630, and rs376038796 were highlighted as potentially harmful, likely to be damaging, and associated with diseases. Further analysis using I-Mutant2.0 and MUpro suggested that the stability of the CTLA4 protein could be altered by these nsSNPs. Additionally, ConSurf analysis indicated high conservation across most regions of the protein, suggesting that these mutations may significantly alter the protein's physico-chemical characteristics, such as size, charge, and hydrophobicity. These changes, in turn, could impact the protein's stability and function, potentially leading to disease.

The results of this study suggest that the R8Q, R8W, P32S, and A86V mutations may contribute to the destabilization of the protein, which has also been confirmed by MD simulation. Finally, PCA study also reveals the genetic differences between wild-type and four SNP due to mutation. However, it is important to acknowledge that these conclusions are based on *in silico* analysis, and the potential link between these nsSNPs and disease susceptibility across diverse populations requires further experimental validation.

## Supporting information

**S1 Table. KEGG pathway enrichment analysis of CTLA4 gene and its associate gene.**
(DOCX)

**S2 Table. Number of hydrogen bonds analysis of wild type and R8Q, R8W, P32S and A86V mutant type protein over 100 ns simulation.**
(XLSX)

## Acknowledgments

AKM Munzurul Hasan is the recipient of University Graduate Scholarships (UGS) and Graduate Teaching Fellowships (GTF) at the University of Saskatchewan.

## Author Contributions

**Conceptualization:** Md. Motiar Rahman, Md. Munnaf Hossen.

**Formal analysis:** Md. Mostafa Kamal, Shamiha Tabassum Teeya, Tanveer A. Wani, Atekah Hazzaa Alshammari.

**Funding acquisition:** Douglas P. Chivers.

**Investigation:** Md. Motiar Rahman, Francisco Carlos da Silva Junior, Md. Munnaf Hossen.

**Methodology:** Kazi Fahmida Haque Shantanu, Shamiha Tabassum Teeya, Md. Munnaf Hossen.

**Software:** Tanveer A. Wani, Atekah Hazzaa Alshammari.

**Supervision:** A. K. M. Munzurul Hasan, Md. Munnaf Hossen.

**Validation:** Francisco Carlos da Silva Junior, Md. Munnaf Hossen.

**Visualization:** Md. Munnaf Hossen.

**Writing – original draft:** Md. Mostafa Kamal, Kazi Fahmida Haque Shantanu, Md. Motiar Rahman, A. K. M. Munzurul Hasan, Md. Munnaf Hossen.

**Writing – review & editing:** A. K. M. Munzurul Hasan, Douglas P. Chivers, Mahesh Rachamalla, Francisco Carlos da Silva Junior, Md. Munnaf Hossen.

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
