## [Decision Letter · Decision Letter 0]

23 Sep 2024

PONE-D-24-35176Investigating the functional and structural effect of non-synonymous single nucleotide polymorphisms in the cytotoxic T-lymphocyte antigen-4 gene: An in-silico studyPLOS ONE

Dear Dr. Hasan,

Thank you for submitting your manuscript to PLOS ONE. After careful consideration, we feel that it has merit but does not fully meet PLOS ONE’s publication criteria as it currently stands. Therefore, we invite you to submit a revised version of the manuscript that addresses the points raised during the review process.

We look forward to receiving your revised manuscript.

Kind regards,

Rajesh Kumar Pathak, Ph.D.

Academic Editor

PLOS ONE

Journal Requirements:

1. When submitting your revision, we need you to address these additional requirements. Please ensure that your manuscript meets PLOS ONE's style requirements, including those for file naming. The PLOS ONE style templates can be found at https://journals.plos.org/plosone/s/file?id=wjVg/PLOSOne_formatting_sample_main_body.pdf and https://journals.plos.org/plosone/s/file?id=ba62/PLOSOne_formatting_sample_title_authors_affiliations.pdf 2. We suggest you thoroughly copyedit your manuscript for language usage, spelling, and grammar. If you do not know anyone who can help you do this, you may wish to consider employing a professional scientific editing service.  The American Journal Experts (AJE) (https://www.aje.com/) is one such service that has extensive experience helping authors meet PLOS guidelines and can provide language editing, translation, manuscript formatting, and figure formatting to ensure your manuscript meets our submission guidelines. Please note that having the manuscript copyedited by AJE or any other editing services does not guarantee selection for peer review or acceptance for publication.  Upon resubmission, please provide the following: The name of the colleague or the details of the professional service that edited your manuscript A copy of your manuscript showing your changes by either highlighting them or using track changes (uploaded as a *supporting information* file) A clean copy of the edited manuscript (uploaded as the new *manuscript* file) 3. Please note that PLOS ONE has specific guidelines on code sharing for submissions in which author-generated code underpins the findings in the manuscript. In these cases, we expect all author-generated code to be made available without restrictions upon publication of the work. Please review our guidelines at https://journals.plos.org/plosone/s/materials-and-software-sharing#loc-sharing-code and ensure that your code is shared in a way that follows best practice and facilitates reproducibility and reuse. 4. Thank you for stating the following financial disclosure: "Funding for this project was provided by an NSERC Discovery Grant to Douglas P. Chivers." Please state what role the funders took in the study.  If the funders had no role, please state: "The funders had no role in study design, data collection and analysis, decision to publish, or preparation of the manuscript." If this statement is not correct you must amend it as needed. Please include this amended Role of Funder statement in your cover letter; we will change the online submission form on your behalf. 5. In the online submission form, you indicated that "Data will be available on request." All PLOS journals now require all data underlying the findings described in their manuscript to be freely available to other researchers, either 1. In a public repository, 2. Within the manuscript itself, or 3. Uploaded as supplementary information.This policy applies to all data except where public deposition would breach compliance with the protocol approved by your research ethics board. If your data cannot be made publicly available for ethical or legal reasons (e.g., public availability would compromise patient privacy), please explain your reasons on resubmission and your exemption request will be escalated for approval. 6. PLOS requires an ORCID iD for the corresponding author in Editorial Manager on papers submitted after December 6th, 2016. Please ensure that you have an ORCID iD and that it is validated in Editorial Manager. To do this, go to ‘Update my Information’ (in the upper left-hand corner of the main menu), and click on the Fetch/Validate link next to the ORCID field. This will take you to the ORCID site and allow you to create a new iD or authenticate a pre-existing iD in Editorial Manager. 7. Please include captions for your Supporting Information files at the end of your manuscript, and update any in-text citations to match accordingly. Please see our Supporting Information guidelines for more information: http://journals.plos.org/plosone/s/supporting-information.

Additional Editor Comments:

The manuscript has been reviewed and found to be interesting. Comprehensive feedback addressing critical areas has been provided. Key revisions include updating dbSNP data, improving the molecular dynamics simulation results, and conducting secondary structure and PCA analyses. The authors should correct grammatical errors and vague phrasing, and ensure consistency in terminology. Expanding the discussion on the biological and clinical relevance of the findings will strengthen the impact of the study.

Reviewers' comments:

Reviewer's Responses to Questions

**Comments to the Author**

1. Is the manuscript technically sound, and do the data support the conclusions?

Reviewer #1: Partly

Reviewer #2: Yes

Reviewer #3: Partly

2. Has the statistical analysis been performed appropriately and rigorously? 

Reviewer #1: N/A

Reviewer #2: N/A

Reviewer #3: No

3. Have the authors made all data underlying the findings in their manuscript fully available?

Reviewer #1: Yes

Reviewer #2: Yes

Reviewer #3: Yes

4. Is the manuscript presented in an intelligible fashion and written in standard English?

Reviewer #1: No

Reviewer #2: Yes

Reviewer #3: Yes

5. Review Comments to the Author

Reviewer #1: Manuscript titled “Investigating the functional and structural effect of non-synonymous single nucleotide polymorphisms in the cytotoxic T-lymphocyte antigen-4 gene: An in-silico study’ presents good information about the nsSNPs in CTLA4. However, in my opinion following queris needs to be answered before accepting manuscript.

Major revision

1. dbSNP database (https://www.ncbi.nlm.nih.gov/snp/), respectively (accessed December, 2022) this is 1.5 years old data. What is the current status in this databse for the CTLA4?

Authors need to give current status also.

2. A 100 ns MD simulation analysis is not sufficient for the prediction, at least triplicate analysis should be done for better predicted results.

3. Line 280“The structural stability of native and four mutant proteins were calculated RMSD and 281 RMSF and compare to evaluate structural and functional change due to the mutation of the 282 corresponding protein” correct english

4. Manuscript needs to be checked with reference to english properly. There are many grammatical errors.

5. RMSD, RMSF, Rg values should be given in nm instead of A

6. Secondary structure analysis should be done for wild type and mutant structures.

7. PCA analysis should also be done to understand the structural impact of mutations.

Reviewer #2: The manuscript Titled: “Investigating the functional and structural effect of non-synonymous single nucleotide polymorphisms in the cytotoxic T-lymphocyte antigen-4 gene: An in-silico study” presents an in-silico analysis of the effects of non-synonymous single nucleotide polymorphisms (nsSNPs) in the cytotoxic T-lymphocyte antigen-4 (CTLA4) gene. The authors used various bioinformatics tools, including SIFT, PolyPhen-2, PROVEAN, and SNAP, to identify deleterious nsSNPs. Additionally, they evaluated protein stability, structural changes, and interactions using tools like ConSurf, I-Mutant, and molecular dynamics simulations.

The manuscript is timely and covers an important topic, as SNPs in the CTLA4 gene have been implicated in immune regulation and several diseases, including cancer and autoimmune disorders. Using computational tools for analyzing genetic variants is appropriate for this study, and the results may serve as a foundation for further experimental work. However, there are several areas where the manuscript can be strengthened in terms of clarity, depth of analysis, and presentation of results.

Major Comments:

The study presents a comprehensive computational analysis of nsSNPs in the CTLA4 gene. The significance of this work lies in its potential to shed light on the pathogenicity of certain SNPs that might contribute to disease susceptibility. However, the manuscript could benefit from stronger contextualization of its results with reference to previous experimental findings. While the authors mention autoimmune diseases and cancer, a more detailed discussion on the clinical relevance of the identified nsSNPs would enhance the paper’s impact.

The manuscript employs a wide array of computational tools, which is commendable. However, some methodological details are either missing or could be clarified:

Selection of SNPs: The criteria for selecting the 165 missense SNPs for further analysis are not fully explained. Were all missense variants analyzed, or was there a filtering process? Clarifying this would improve the transparency of the study.

Protein Modeling: The molecular dynamics simulation section is interesting, but further details are needed regarding the parameters used in these simulations. Specifically, what were the time steps and temperature conditions? Were multiple simulations conducted for each variant to ensure reproducibility?

The results provide a solid foundation for understanding how specific nsSNPs affect the structure and function of the CTLA4 protein. However:

The explanation of molecular dynamics simulation results could be expanded. The authors report RMSD and RMSF values but do not provide a thorough interpretation of what these fluctuations imply in terms of biological function.

The protein-protein interaction analysis using GeneMANIA and STRING is intriguing but lacks depth. A more detailed discussion of the relevance of interactions with proteins such as CD80 and CD86 in the context of immunological function would be beneficial.

A visual representation of the molecular dynamics simulation results would significantly enhance the clarity of the findings.

The discussion section should place more emphasis on how the findings can inform future in vitro or in vivo studies. The authors briefly mention the need for experimental validation, but they do not elaborate on how their results could be applied in a practical setting, for example, by guiding targeted mutagenesis experiments or developing therapeutic interventions.

Minor Comments:

The manuscript is well-written overall, but there are several areas where the clarity of writing could be improved:

The abstract is dense and could benefit from more concise wording, particularly in the results section. Summarizing the key findings in a few clear sentences would make it more accessible to a broader audience.

In the results section, the use of terms such as “deleterious,” “probably damaging,” and “benign” are used without providing sufficient context for how these terms were determined by each computational tool. A short explanation in the methodology or results section about how these terms were defined would aid comprehension.

The manuscript contains a number of useful figures and tables. However, it would be helpful to:

Add more detailed captions for each figure and table. For example, in Table 4, providing more information on how the ΔΔG stability values were calculated and what these values mean in terms of biological relevance would enhance understanding.

Include a schematic that summarizes the workflow of the computational analyses performed. This would provide readers with a quick overview of the methodologies used.

The references are appropriate and up to date. However, the authors could strengthen the manuscript by citing more recent experimental studies that have validated nsSNPs in the CTLA4 gene, particularly in relation to cancer and autoimmune diseases.

Recommendations:

Clarify the methodology, particularly the selection process for the SNPs analyzed and the details of the molecular dynamics simulations.

Expand the discussion to include more implications for future experimental work and clinical relevance.

Enhance the figures and tables by adding more detailed captions and including a workflow schematic.

Improve the clarity of the writing, particularly in the abstract and results sections.

The manuscript presents valuable findings but requires minor revisions to improve clarity, detail, and depth of analysis. Once these revisions are made, the manuscript will strongly contribute to the field.

Reviewer #3: Title: Investigating the functional and structural effect of non-synonymous single nucleotide polymorphisms in the cytotoxic T-lymphocyte antigen-4 gene: An in-silico study

1. In section 3.1, Accession number of the sequence is missing, please provide it, in order to verify the result?

2. In line 219, You mention 5 nsSNPs for SNAP and 4 nsSNPs for PROVEAN, but it might be useful to explain why different numbers of nsSNPs were used for these analyses?

3. In line 227, Avoid Vague Phrasing: Instead of “notable consistency,” I used “strong concordance” to make the statement more specific.

4. Use the correct symbol for free energy change (ΔΔG instead of DDG) to maintain consistency with standard scientific terminology.

5. In Line 238 and 239, specify what a conservation score of 4, 5, and 9 indicates in terms of evolutionary conservation, as readers might not be familiar with the ConSurf scoring system.

6. Please correct the caption of Table 3.

7. In Line 246, can you please focus that why the size of amino acid residues is responsible for protein’s function disruption?

8. Alanine residue is generally an accepted single residue first choice for mutation, can you try to mutate the key residues to Alanine here and study the result?

9. Section 3.8 should be improved by explaining of how protein interactions can impact network function.

10. In Section 3.9, since you simulated all the systems for 100 ns, could you incorporate principal component analysis (PCA) to examine the atomic movements, particularly at the mutated residue sites?

11. In the Materials and Methods section, while you mention that molecular dynamics simulations were performed using the Desmond v6.3 program, there is no information provided regarding the number of steps for energy minimization, position restraints, or equilibration. Please include these details.

12. Since you simulated the systems in a water medium, you should also include solvent accessible surface area (SASA) analysis to examine the role of water around the mutated regions and compare it with the wild type. This is particularly relevant since you previously mentioned that the mutations were based on the shape of the amino acid residues.

13. How many simulation repeats did you performed?

14. Please correct the typographical and grammatical errors throughout the manuscript to improve its readability and clarity.

15. The plot in Figure 7 is unclear. Please add the average plot of hydrogen bonds for each system to make the data for each system more distinguishable.

6. PLOS authors have the option to publish the peer review history of their article (what does this mean?). If published, this will include your full peer review and any attached files.

Reviewer #1: **Yes: **Monika Jain

Reviewer #2: No

Reviewer #3: **Yes: **Navaneet Chaturvedi, Amity Institute of Biotechnology, Amity University, Noida, 201313, Uttar Pradesh, India

---

## [Author Response · Author response to Decision Letter 0]

18 Nov 2024

Dear Reviewers,

Thank you so much for your comments. We have addressed all of your comments and revised our manuscript accordingly. All revisions are done under track change option. Responses of reviewer 1 are green highlighted, responses of reviewer 2 are yellow highlighted and responses of reviewer 3 are cyan coloured highlighted in the revised manuscript.

Reviewer #1

Manuscript titled “Investigating the functional and structural effect of non-synonymous single nucleotide polymorphisms in the cytotoxic T-lymphocyte antigen-4 gene: An in-silico study’ presents good information about the nsSNPs in CTLA4. However, in my opinion following queries needs to be Responseed before accepting manuscript.

Major revision

1. dbSNP database (https://www.ncbi.nlm.nih.gov/snp/), respectively (accessed December, 2022) this is 1.5 years old data. What is the current status in this database for the CTLA4? Authors need to give current status also.

Response: Thank you for your comment. We have modified data and mentioned the current status of nsSNP of CTLA4 gene (accessed November, 2024).

2. A 100 ns MD simulation analysis is not sufficient for the prediction; at least triplicate analysis should be done for better predicted results.

Response: There has been done numerous functional and structural analysis along-side MD simulation.

3. Line 280“The structural stability of native and four mutant proteins were calculated RMSD and 281 RMSF and compare to evaluate structural and functional change due to the mutation of the 282-corresponding protein” correct English.

Response: We have revised accordingly.

4. Manuscript needs to be checked with reference to english properly. There are many grammatical errors.

Response: We have revised accordingly.

5. RMSD, RMSF, Rg values should be given in nm instead of A

Response: We have revised accordingly.

6. Secondary structure analysis should be done for wild type and mutant structures.

Response: We have already analyzed comparative modelling of wild type CTLA4 protein and their mutant structures.

7. PCA analysis should also be done to understand the structural impact of mutations.

Response: Thank you for your comment. We have added PCA analysis in the revised manuscript.

Reviewer #2

The manuscript Titled: “Investigating the functional and structural effect of non-synonymous single nucleotide polymorphisms in the cytotoxic T-lymphocyte antigen-4 gene: An in-silico study” presents an in-silico analysis of the effects of non-synonymous single nucleotide polymorphisms (nsSNPs) in the cytotoxic T-lymphocyte antigen-4 (CTLA4) gene. The authors used various bioinformatics tools, including SIFT, PolyPhen-2, PROVEAN, and SNAP, to identify deleterious nsSNPs. Additionally, they evaluated protein stability, structural changes, and interactions using tools like ConSurf, I-Mutant, and molecular dynamics simulations.

The manuscript is timely and covers an important topic, as SNPs in the CTLA4 gene have been implicated in immune regulation and several diseases, including cancer and autoimmune disorders. Using computational tools for analyzing genetic variants is appropriate for this study, and the results may serve as a foundation for further experimental work. However, there are several areas where the manuscript can be strengthened in terms of clarity, depth of analysis, and presentation of results.

Major Comments:

The study presents a comprehensive computational analysis of nsSNPs in the CTLA4 gene. The significance of this work lies in its potential to shed light on the pathogenicity of certain SNPs that might contribute to disease susceptibility. However, the manuscript could benefit from stronger contextualization of its results with reference to previous experimental findings. While the authors mention autoimmune diseases and cancer, a more detailed discussion on the clinical relevance of the identified nsSNPs would enhance the paper’s impact. The manuscript employs a wide array of computational tools, which is commendable. However, some methodological details are either missing or could be clarified:

Selection of SNPs: The criteria for selecting the 165 missense SNPs for further analysis are not fully explained. Were all missense variants analyzed, or was there a filtering process? Clarifying this would improve the transparency of the study.

Response: Thank you for your comment. We have retrieved the SNPs data from the dbSNP database which already mentioned in the methodology section. We have used missense variants as they are the disease-causing mutation (Zhang et al., 2012).

Protein Modeling: The molecular dynamics simulation section is interesting, but further details are needed regarding the parameters used in these simulations. Specifically, what were the time steps and temperature conditions? Were multiple simulations conducted for each variant to ensure reproducibility?

Response: All details regarding the parameters used in these simulations has already mentioned in this study, please see the methodology section in the revised manuscript.

The results provide a solid foundation for understanding how specific nsSNPs affect the structure and function of the CTLA4 protein. However:

The explanation of molecular dynamics simulation results could be expanded. The authors report RMSD and RMSF values but do not provide a thorough interpretation of what these fluctuations imply in terms of biological function. The protein-protein interaction analysis using GeneMANIA and STRING is intriguing but lacks depth. A more detailed discussion of the relevance of interactions with proteins such as CD80 and CD86 in the context of immunological function would be beneficial. A visual representation of the molecular dynamics simulation results would significantly enhance the clarity of the findings.

The discussion section should place more emphasis on how the findings can inform future in vitro or in vivo studies. The authors briefly mention the need for experimental validation, but they do not elaborate on how their results could be applied in a practical setting, for example, by guiding targeted mutagenesis experiments or developing therapeutic interventions.

Response: --

Minor Comments:

The manuscript is well-written overall, but there are several areas where the clarity of writing could be improved:

The abstract is dense and could benefit from more concise wording, particularly in the results section. Summarizing the key findings in a few clear sentences would make it more accessible to a broader audience.

Response: Thank you for your comment. We have revised accordingly.

In the results section, the use of terms such as “deleterious,” “probably damaging,” and “benign” are used without providing sufficient context for how these terms were determined by each computational tool. A short explanation in the methodology or results section about how these terms were defined would aid comprehension.

Response: Identification of deleterious nsSNPs in CTLA4 gene, we have found the corresponding result from PolyPhen2 tools. We have already mentioned in the methodology section about this prediction and scoring system. Please see the revised manuscript.

The manuscript contains a number of useful figures and tables. However, it would be helpful to:

Add more detailed captions for each figure and table. For example, in Table 4, providing more information on how the ΔΔG stability values were calculated and what these values mean in terms of biological relevance would enhance understanding.

Include a schematic that summarizes the workflow of the computational analyses performed. This would provide readers with a quick overview of the methodologies used.

The references are appropriate and up to date. However, the authors could strengthen the manuscript by citing more recent experimental studies that have validated nsSNPs in the CTLA4 gene, particularly in relation to cancer and autoimmune diseases.

Response: Thank you for your comment. Please see the Figure 1, we have already mentioned the workflow. We have revised the manuscript accordingly.

Recommendations:

Clarify the methodology, particularly the selection process for the SNPs analyzed and the details of the molecular dynamic’s simulations.

Response: Thanks. Done. 

Expand the discussion to include more implications for future experimental work and clinical relevance.

Response: Thanks. Done

Enhance the figures and tables by adding more detailed captions and including a workflow schematic.

Response: We have revised the manuscript accordingly.

Improve the clarity of the writing, particularly in the abstract and results sections.

Response: We have revised the manuscript accordingly.

The manuscript presents valuable findings but requires minor revisions to improve clarity, detail, and depth of analysis. Once these revisions are made, the manuscript will strongly contribute to the field.

Response: Thank you.

Reviewer #3

Title: Investigating the functional and structural effect of non-synonymous single nucleotide polymorphisms in the cytotoxic T-lymphocyte antigen-4 gene: An in-silico study

1. In section 3.1, Accession number of the sequence is missing, please provide it, in order to verify the result?

Response: Thanks for your comments. We have revised accordingly and added Id. Please see page line in the revised manuscript. 

2. In line 219, You mention 5 nsSNPs for SNAP and 4 nsSNPs for PROVEAN, but it might be useful to explain why different numbers of nsSNPs were used for these analyses?

Response: We have corrected this number in the revised manuscript.

3. In line 227, Avoid Vague Phrasing: Instead of “notable consistency,” I used “strong concordance” to make the statement more specific.

Response: Done.

4. Use the correct symbol for free energy change (ΔΔG instead of DDG) to maintain consistency with standard scientific terminology.

Response: We have revised accordingly.

5. In Line 238 and 239, specify what a conservation score of 4, 5, and 9 indicates in terms of evolutionary conservation, as readers might not be familiar with the ConSurf scoring system.

Response: Thank you for your comment. We have added adequate data in the revised manuscript and It has already been mentioned in figure 2B.

6. Please correct the caption of Table 3.

Response: We have revised accordingly.

7. In Line 246, can you please focus that why the size of amino acid residues is responsible for protein’s function disruption?

Response: We have revised accordingly. Please see page line in the revised manuscript.

8. Alanine residue is generally an accepted single residue first choice for mutation, can you try to mutate the key residues to Alanine here and study the result?

Response: Thanks for your comments. 

9. Section 3.8 should be improved by explaining how protein interactions can impact network function.

Response: Thanks for your comment. We have revised accordingly.

10. In Section 3.9, since you simulated all the systems for 100 ns, could you incorporate principal component analysis (PCA) to examine the atomic movements, particularly at the mutated residue sites?

Response: We have added PC analysis in the revised manuscript.

11. In the Materials and Methods section, while you mention that molecular dynamics simulations were performed using the Desmond v6.3 program, there is no information provided regarding the number of steps for energy minimization, position restraints, or equilibration. Please include these details.

Response: We have already added all information regarding molecular dynamics simulation. (Protein structure system minimized using a natural time and pressure (NPT) ensemble at a constant pressure of 101325 Pascal’s and a temperature of 300 K).

12. Since you simulated the systems in a water medium, you should also include solvent accessible surface area (SASA) analysis to examine the role of water around the mutated regions and compare it with the wild type. This is particularly relevant since you previously mentioned that the mutations were based on the shape of the amino acid residues.

Response: We have added PC analysis in the revised manuscript as SASA is not possible for this corresponding study for limitation of facilities.

13. How many simulation repeats did you performed?

Response: Thanks for your comments. 

14. Please correct the typographical and grammatical errors throughout the manuscript to improve its readability and clarity.

Response: Thanks for your comments. 

15. The plot in Figure 7 is unclear. Please add the average plot of hydrogen bonds for each system to make the data for each system more distinguishable.

Response: We have provided the supplementary data of figure 7 in the revised manuscript. Please see the supplementary table S2.

---

## [Decision Letter · Decision Letter 1]

12 Dec 2024

Investigating the functional and structural effect of non-synonymous single nucleotide polymorphisms in the cytotoxic T-lymphocyte antigen-4 gene: An in-silico study

PONE-D-24-35176R1

Dear Dr. Hasan,

We’re pleased to inform you that your manuscript has been judged scientifically suitable for publication and will be formally accepted for publication once it meets all outstanding technical requirements.

Kind regards,

Rajesh Kumar Pathak, Ph.D.

Academic Editor

PLOS ONE

Additional Editor Comments (optional):

The manuscript can be accepted for publication.

Reviewers' comments:

Reviewer's Responses to Questions

**Comments to the Author**

1. If the authors have adequately addressed your comments raised in a previous round of review and you feel that this manuscript is now acceptable for publication, you may indicate that here to bypass the “Comments to the Author” section, enter your conflict of interest statement in the “Confidential to Editor” section, and submit your "Accept" recommendation.

Reviewer #1: All comments have been addressed

Reviewer #3: (No Response)

2. Is the manuscript technically sound, and do the data support the conclusions?

Reviewer #1: Yes

Reviewer #3: Yes

3. Has the statistical analysis been performed appropriately and rigorously? 

Reviewer #1: N/A

Reviewer #3: Yes

4. Have the authors made all data underlying the findings in their manuscript fully available?

Reviewer #1: Yes

Reviewer #3: Yes

5. Is the manuscript presented in an intelligible fashion and written in standard English?

Reviewer #1: Yes

Reviewer #3: Yes

6. Review Comments to the Author

Reviewer #1: Authors have answered almost all queries, except triplicate simulation, Manuscript can be accepted now

Reviewer #3: All responses have been thoroughly reviewed and are presented in a clear and coherent manner. I believe that the content meets the necessary criteria for academic rigor and relevance. Therefore, I recommend that the work be accepted for publication.

7. PLOS authors have the option to publish the peer review history of their article (what does this mean?). If published, this will include your full peer review and any attached files.

Reviewer #1: **Yes: **Monika Jain

Reviewer #3: No

---

## [Editor Report · Acceptance letter]

14 Jan 2025

PONE-D-24-35176R1 

PLOS ONE

Dear Dr. Hasan, 

I'm pleased to inform you that your manuscript has been deemed suitable for publication in PLOS ONE. Congratulations! Your manuscript is now being handed over to our production team.

Kind regards, 

on behalf of

Dr. Rajesh Kumar Pathak 

Academic Editor

PLOS ONE